# Elastic Bioresorbable Polymeric Capsules for Osmosis-Driven Delayed Burst Delivery of Vaccines

**DOI:** 10.3390/pharmaceutics13030434

**Published:** 2021-03-23

**Authors:** Kerr D. G. Samson, Eleonore C. L. Bolle, Mariah Sarwat, Tim R. Dargaville, Ferry P. W. Melchels

**Affiliations:** 1Institute of Biological Chemistry, Biophysics and Bioengineering, School of Engineering and Physical Sciences, Heriot-Watt University, Edinburgh EH14 4AS, UK; k.samson@hw.ac.uk; 2Institute of Health and Biomedical Innovation, Queensland University of Technology, Brisbane, QLD 4059, Australia; e.bolle@uq.edu.au (E.C.L.B.); mariah.sarwat@hdr.qut.edu.au (M.S.); t.dargaville@qut.edu.au (T.R.D.)

**Keywords:** biodegradable polymers, delayed release, burst release, osmosis, vaccine delivery

## Abstract

Single-administration vaccine delivery systems are intended to improve the efficiency and efficacy of immunisation programs in both human and veterinary medicine. In this work, an osmotically triggered delayed delivery device was developed that was able to release a payload after a delay of approximately 21 days, in a consistent and reproducible manner. The device was constructed out of a flexible poly(ε-caprolactone) photo-cured network fabricated into a hollow tubular shape, which expelled approximately 10% of its total payload within 2 days after bursting. Characterisation of the factors that control the delay of release demonstrated that it was advantageous to adjust material permeability and device wall thickness over manipulation of the osmogent concentration in order to maintain reproducibility in burst delay times. The photo-cured poly(ε-caprolactone) network was shown to be fully degradable in vitro, and there was no evidence of cytotoxicity after 11 days of direct contact with primary dermal fibroblasts. This study provides strong evidence to support further development of flexible biomaterials with the aim of continuing improvement of the device burst characteristics in order to provide the greatest chance of the devices succeeding with in vivo vaccine booster delivery.

## 1. Introduction

Immunisation is regarded as one of the most successful and cost-effective forms of medical intervention, with vaccinations preventing an estimated 2.5 million deaths worldwide and saving millions of dollars in health care costs per year [1,2,3,4]. Access to common vaccines is regarded by the World Health Organisation (WHO) as a critical component of a well-functioning health care system, and the availability of immunisation is classed as a core component of the human right to health [1,5]. Vaccines produce their prophylactic effect by inducing a pathogen-specific immune response to an antigen that primes the body’s immune system and generates lasting immunological memory, thus enabling an efficient secondary immune response when the stimulus is encountered again in the future [6].

One of the challenges with the standard injection administration method is the need for subsequent booster doses at specific time intervals following the original dose. This need for multiple single bolus injections can have a negative effect on patient compliance, and creates logistical issues which are greatly exacerbated in developing countries [7]. Alternatives to multiple vaccine administration is also of interest to the companion animal, cattle, sheep and wildlife markets, where vaccination programmes can be expensive and labour intensive [8]. In the case of wildlife [9], the animal may need to be kept in captivity or recaptured to administer future doses, introducing further expense and complications.

Due to these shortfalls, there is great interest in development of single-administration vaccine (SAV) delivery devices and other administration methods, which are recognised as an important part of the WHO’s Global Vaccine Action Plan [1]. Development of non-injectable delivery routes such as oral and intranasal delivery have been attempted, but only a limited few have shown clinical and regulatory success to date [10,11,12]. Most of the proposed SAV solutions employ resorbable biomaterials to delay the release of (a part of) the antigen, e.g., in the form of injectable microparticles [13,14] or transdermal microneedle arrays [15,16]. We previously introduced an alternative approach, using an implantable monolithic polymer capsule that contains the vaccine dose and an osmotically active agent [17]. The internal osmogent results in a slow influx of water that produces an increasing hydrostatic pressure within the device that ultimately overcomes the burst pressure of the capsule, releasing the contents. This approach offers several advantages over contemporary methods, at the cost of ease of administration. The proof-of-principle study employed dip-coated tubes of thermoplastic poly(ε-caprolactone) (PCL), demonstrating in vitro release of a model dye after a delay of up to 11 days [17]. Besides the delay time being too short for most SAV applications, another limitation was the incomplete release at burst.

A similar delayed release concept has been presented in patent literature [18], where a swellable osmotic core was coated with multiple films to delay and control water uptake. During in vitro testing these devices demonstrated delayed rupture by a period of between 13 and 67 days, producing a pulsatile release profile. Devices loaded with *C. botulinum* C toxoid were compared against a bolus liquid injection of the same payload in an in vivo study in mice. The mice in the device group displayed delayed seroconversion by 38 days, compared with a 14 day delay from the injection group. However, this patent dates back to 1992 and the last reference to this device was within a review by Stubbe et al. in 2004 [19], leaving the current development status unknown.

Despite the idea of employing osmosis to trigger burst release being present in the patent literature for over two decades, it appears to have remained relatively unexplored. This is both in comparison to other mechanisms for delayed burst delivery, as well is in comparison to methods of exploiting osmosis for sustained release, such as zero-order release osmotic pumps. Continuous-release osmotic pumps have been developed over the last 40 years for use as oral dosage forms or subcutaneous or intraperitoneal implants. The osmotic-controlled release oral delivery systems (OROS) are currently marketed for a wide variety of pharmacotherapies covering various fields such as cardiology, endocrinology, and urology [20]. The implantable forms developed by the ALZA corporation are manufactured from non-degradable materials [21] and are currently only licensed for experimental animal use (not animals used for food products) [22]. However, the leuprolide acetate delivery implant marketed as Viadur^®^ by Bayer was approved by the FDA for human use to manage the symptoms of advanced prostate cancer in 2000, but was discontinued in 2007 due to diminished market demand and rising manufacturing costs [23]. From these studies and applications, it is known that osmosis can be used as a reliable trigger for drug release, showing little sensitivity to changes in environmental conditions, as well as high in vitro to in vivo correlation [24].

The current study builds on our previous work, based on the hypothesis that devices fabricated from a polymer with increased elasticity will release more payload at the moment of burst. In this work, we introduced crosslinks to PCL, expecting that this will provide a greater degree of elastic shape recovery than for the non-crosslinked PCL material. If devices return, at least partially, to their original shape, they will eject more payload in the process.

Crosslinked PCL (xPCL) devices were prepared by dip coating methacrylated PCL oligomers, followed by photocuring. Their osmosis-driven burst delivery behaviour is assessed in a head-to-head comparison with devices prepared from non-crosslinked PCL, which are almost identical to those reported before [17]. Additionally, many improvements over the previous study are presented, including automated fabrication, and persistent monitoring of mass, and of released dye through UV spectrophotometry. Furthermore, we evaluated the suitability of using the initial osmogent loading concentration to tailor the delay time. Devices with a higher initial loading concentration of osmogent are expected to feature a greater rate of water uptake and will therefore burst quicker than devices with lower osmogent loading concentrations, but a decrease in driving force may also result in an increase in inaccuracies in the resulting delay time.

## 2. Materials and Methods

### 2.1. Materials

Polycaprolactone (PCL) diol (M_n_ = 10,000 g/mol), PCL triol (M_n_ = 900 g/mol), PCL (M_n_ = 45,000 g/mol), methacrylic anhydride (MAAh), sodium hydroxide pellets (NaOH), and magnesium sulphate (MgSO_4_) were acquired from Sigma Aldrich (Gillingham, UK); potassium carbonate (K_2_CO_3_), toluene, dichloromethane (DCM), and hexane were obtained from Fisher Scientific (Loughborough, UK). PURASORB^®^ PC12 medical grade PCL (70,000 g/mol) was purchased from Corbion (Amsterdam, The Netherlands). Benzyl alcohol was obtained from Alfa Aesar (Heysham, UK). Diphenyl (2,4,6-trimethylbenzoyl) phosphine oxide (TPO) was purchased from Sigma Aldrich (UK). Queen branded blue food dye (Brilliant blue FCF) was purchased from Woolworths supermarkets Australia. Ethical approval for human skin collection was obtained from the Queensland University of Technology Research Ethics Committee (1300000063) and the Uniting Healthcare / St Andrew’s Hospital Ethics Committee (0346). Trypsin, Collagenase-A, Dulbecco’s Modified Eagle’s Medium (DMEM), foetal calf serum (FCS), penicillin, streptomycin, l-glutamine, and tetramethylrhodaminyl-conjugated phalloidin (TRITC Phalloidin) and phosphate buffered saline (PBS) were acquired from Invitrogen (Scoresby, VIC, Australia). 4’6-diamidino–2–phenylindole (DAPI) was acquired from Life Technologies (Waltham, MA, USA). PBS tablets, cell proliferation reagent WST-1, Triton X-100, and bovine serum albumin (BSA) were purchased from Sigma Aldrich (Castle Hill, NSW, Australia). Paraformaldehyde (PFA) was purchased from ProSciTech (Kirwan, QLD, Australia). Human dermal fibroblasts were isolated from split thickness biopsies from consenting patients undergoing abdominoplasty and breast reduction surgery [25], according to previously published protocols [26]. Human dermal fibroblasts were cultured in Dulbecco’s Modified Eagle’s Medium (DMEM) supplemented with 10% FCS, 2 mM L-glutamine, 50 U/mL of penicillin and 50 μg/mL of streptomycin.

### 2.2. Polymers

The oligomers were functionalised by reacting the terminal hydroxyl groups with methacrylic anhydride under a nitrogen atmosphere at 130 °C in the presence of the proton scavenger K_2_CO_3_. Due to the increased viscosity of the 10,000 g/mol oligomer, small quantities of anhydrous toluene were added until smooth stirring was achieved. A molar excess of 50–100 mol% MAAh and K_2_CO_3_ was used. Proton-nuclear magnetic resonance spectrophotometry (^1^H-NMR, CDCl_3_, Bruker AVIII 300 MHz) was used to determine the degree of functionalisation of the macromers. The reaction was continued until a high degree of functionalisation (99%+) was reached.

The 10,000 g/mol macromer solution was precipitated into cold (−80 °C) hexane and filtered. The yield was 80–90%. The 900 g/mol macromer formed a low viscosity wax upon precipitation in cold hexane. The upper phase was decanted and discarded, then more hexane was added, and the contents mixed vigorously. This process was repeated until a clear upper phase was formed. The wax was then transferred to suitable centrifugation containers and mixed with DCM and 2 M NaOH solution in a 1:1:6 volume ratio. The solutions were subjected to 2000 g for 5 min at 25 °C within a Beckman Coulter Allegra X-12R centrifuge. The upper aqueous phase was decanted off and discarded, with the volume replaced with demineralised water. The container was shaken, and then returned to the centrifuge. This process was repeated until a clear aqueous phase was achieved. The DCM/macromer phase was mixed with MgSO_4_ to bind any remaining water, and then filtered to remove the solid MgSO_4_ hydrate. Rotary evaporation was used to remove the DCM, leaving behind the waxy macromer with a yield of ~70%.

### 2.3. Film Tensile Strips and Mechanical Testing

A photo-crosslinkable liquid resin was formulated by mixing 30 wt% 10,000 g/mol macromer with 15 wt% 900 g/mol macromer in 55 wt% benzyl alcohol, with a further overall 2 wt% of TPO photoinitiator. A 10% *w*/*v* polymer solution was prepared by dissolving PCL in chloroform. The non-crosslinked, thermoplastic PCL material will be referred to as PCL, while the crosslinked networks will be denoted as xPCL.

The PCL solution was poured onto a flat sheet of stainless steel within a fume cupboard and spread into a homogeneously thick sheet using an Elcometer 3580 casting knife, which was set to a height of 500 µm. The solvent quickly evaporates, leaving behind a thin semicrystalline sheet of PCL, from which rectangular specimens (5 × 50 mm) were cut using a scalpel. The xPCL strips were formed by casting resin on a microscope slide with the casting knife set at 400 µm. The slide was positioned within an Autodesk EMBER 3D printer. A print file with 5 strips of 5 × 50 mm was loaded to the printer. The resin was illuminated for 60 s, after which the slide was removed, and the strips lifted off gently using a scalpel. Any superficial excess resin was removed from the strips with a paper towel, before post-curing for 90 min (45 min each side) within a UV cabinet (UVP CL-1000 L, 365 nm, 3 mW/cm^2^). The xPCL strips were then extracted in isopropanol for 3 days within Soxhlet apparatus and dried in an oven at 80 °C until achieving a constant mass.

The elastic strain recovery of the PCL and xPCL strips were assessed through a series of incremental strains (1, 5, 10, 20, 140%) applied at 10%/min each followed by a recovery period on a Q800 DMA. Samples were mounted onto the tensile clamp arrangement with a grip-to-grip distance of approximately 10 mm. Each sample was equilibrated at 37 °C before being extended to the target strain. After the target strain was reached, the load was removed from the sample and its recovery was recorded over a 15 min period. The experiment was then repeated with the next target strain using the same sample.

### 2.4. Tubes

Both PCL and xPCL tubes were fabricated using a custom-built automatic dip coating apparatus using the two resins described above. Brass rods (2.0 mm diameter) were repeatedly dipped at a 45° angle into a 15 mL Falcon tube containing the respective polymer solution, while the rods were constantly rotated at approximately 10 rpm. For PCL tubes, the process involved 4 repeated dips, with a 3 min interval of continuous rotation after each dip. After the final dip, an annealing process was performed by positioning the rods vertically in a block in an oven at 70 °C for 1 h. The xPCL coated rods were dipped 3 times and were irradiated with 365 nm light (Omnicure S2000; 65 mW/cm^2^) for 7 min between each dip. After the final dip, a 14 min UV-cure step was performed. The xPCL rods were then moved to a UV cabinet and irradiated for 1 h. After completion of the annealing or post-cure stages, the coated rods were trimmed and submerged in ethanol to facilitate removal of the tube from the rod. The xPCL tubes were extracted as described earlier for the xPCL strips. The tubes were cut to length using only the middle portion of the tubes to ensure homogenous wall thickness.

### 2.5. Osmosis-Driven Release

Tubes for osmotic release experiments were sealed on one end prior to device filling. The PCL devices were capped by thermal sealing as described in Melchels et al. [17]. For the xPCL tubes, 10–15 µL of resin was injected into an open end and cured for 10 min using UV light (Omnicure S2000). Stock solutions of a molality of 3, 3.5 and 4 mol of glucose per kg of solvent (indicated as 3 m, 3.5 m, and 4 m, respectively) were prepared, with the solvent being food dye diluted 4× with distilled water. Devices were loaded with 70 µL of the appropriate glucose/dye stock solution using a syringe fitted with a 22-gauge blunt needle. After filling, the capping process was repeated on the remaining open end.

Filled and sealed devices were submerged within 15 mL Falcon tubes containing 10 mL of PBS. The tubes were kept in an incubator set at 37 °C for the duration of the experiment. Device behaviour was monitored over 72 days by gravimetric and UV spectrophotometry measurements. All measurements were taken every 2–3 days, and devices were inspected visually each time for signs of dye release. Mass measurements were taken on an analytical balance (d = 0.0001 g). A wavelength sweep was performed to find the absorbance maximum of the food dye, which was determined to be at 630 nm and therefore all future absorbance measurements were taken at this wavelength. At each time point, the entire volume of PBS that the device was immersed in was removed and replaced with fresh PBS. A quartz cuvette was filled with a sample of the removed solution. The absorbance at each time point was determined in triplicate and averaged. Blank PBS absorbance readings were deducted from experimental sample absorbance values before being averaged for data analysis. The burst sites of several tubes were imaged using a DinoLite^TM^ Basic AM2111 digital microscope (AnMo Electronics Corporation, Hsinchu City, Taiwan).

### 2.6. Cytotoxicity

PCL disc-shaped specimens were punched from compression moulded films, while xPCL discs were produced by moulding. Samples for cell culture were sterilised by immersion in 80% ethanol for 60 min and dried in a laminar flow cabinet. Following 3 washes in PBS, samples were placed in fibroblast growth medium overnight. Samples for cell proliferation experiments of 5 mm in diameter were placed in a 96 well plate. Isolated fibroblasts, 5 × 10^3^ in 10 µL, were seeded onto the samples and allowed to attach for 1 h, prior to addition of 150 µL fibroblast growth medium. Medium was replaced every 2–3 days. Cellular proliferation on the samples was studied at 1, 4, and 11 days using a WST-1 assay. Upon reaching day 1, 4, and 11, three samples per experimental group were transferred into a 48 well plate, 200 µL of fresh medium was added along with 20 µL of WST-1 reagent. Samples were incubated in the dark for 2 h. The supernatant was then transferred into a 96 well plate and readings taken on a plate reader (BioRad xMark Microplate Spectrophotometer, Bio-Rad Laboratories Inc., Hercules, CA, USA) at 450 nm with the reference wavelength set to 650 nm. All sample group values were expressed as a percent of relative metabolic activity vs. the positive controls.

Samples (12 mm in diameter) to study cellular morphology were seeded with 1.2 × 10^4^ fibroblasts in 10 µL, within a 24 well plate and allowed to attach for 1 h, prior to addition of 1.5 mL of fibroblast growth medium. Medium was replaced every 2–3 days. Cellular density and morphology on the samples was studied at 1, 4, and 11 days using epifluorescence microscopy (Nikon Eclipse Tis, Nikon Corporation, Tokyo, Japan). Upon reaching the time points, the samples for fluorescent microscopy were rinsed in PBS supplemented with calcium and magnesium, and fixed in 4% PFA for one hour at room temperature. Samples were then incubated in 0.2% (*v*/*v*) Triton X-100 in PBS for 5 min, prior to incubation for 10 min in 1% BSA/PBS (*w*/*v*) blocking solution. Samples were fluorescently labelled by placing them in a working solution containing blocking solution, 0.8 U/mL TRITC Phalloidin, and 5 µg/mL DAPI. Samples were incubated in working solution for 45 min on a shaker and protected from light. Stained samples were stored in the dark at 4 °C immersed in PBS until use. Samples were washed 3 times in PBS between each step above.

### 2.7. Statistical Analysis

Statistical analysis of numerical datasets was performed using Minitab 19 software (Minitab LLC, State College, PA, USA). Material elasticity data were assessed through a one-way ANOVA with Games–Howell post hoc, and considered significant if *p* < 0.05. Group means displayed on the figure are assigned a letter (e.g., ‘A’ or ‘A, B’), only groups that share a letter are not significantly different (*p* > 0.05). The water uptake data were also evaluated using a one-way ANOVA with Games–Howell post hoc. Evaluation of the dye release between material groups was performed using a standard 2 sample *t*-test (95% CI). Standard 2 sample *t*-tests were used to evaluate the immediate dye release after burst, dye release rate, and total dye released between the PCL and xPCL groups. A *t*-test was also used to compare each of the polymer material groups against the glass control group for the cytotoxicity data.

## 3. Results

### 3.1. Material Synthesis and Mechanical Properties

Poly(ε-caprolactone) (PCL) macromonomers of high purity (99.9%) and quantitative hydroxyl-to-methacrylate conversion were obtained as confirmed by ^1^H-NMR spectrometry (Appendix A). High- and low-molecular-weight macromers were combined with a solvent and photo-initiator to formulate a liquid resin which was used for the fabrication of crosslinked PCL (xPCL) test samples (the crosslinked network has an essentially infinite molecular weight). Non-crosslinked test samples were also produced using thermoplastic PCL (molecular weight 70 kg/mol). Depiction of the basic structural formulae of the thermoplastic PCL, and both the xPCL macromers, are shown in Appendix A.

The elasticity of the PCL and xPCL materials was compared by assessing the extent of permanent set (permanent plastic deformation) after applying incrementally increasing strains, followed by free recovery (Figure 1). A lower degree of permanent set indicates a greater elastic response.

With the exception of the 1% strain results, the relative amount of permanent set increases when the sample was subjected to greater strain. xPCL samples show a lower degree of permanent set than PCL after all applied target strains (only statistically significant at 10–140% applied strain), with the relative difference increasing when the applied strain is increased from 5% stepwise up to 140%. Additional tensile data for both materials can be found in Appendix A.

### 3.2. Device Manufacture

Thin-walled tubes of xPCL and PCL materials were produced using automated dip-coating apparatus. PCL tubes solidify via solvent evaporation and crystallisation, whereas xPCL solidifies through photo-initiated crosslinking. Analysis by micro-computed tomography of the final tubes revealed high reproducibility in wall thickness between specimens for both tube types, as well as even wall thickness along the length of the tube (Appendix A). PCL tubes had significantly thinner walls than xPCL tubes, with an average ± standard deviation of the wall thickness distribution of representative tubes of 127 ± 13 µm for PCL and 335 ± 29 µm for xPCL. After filling and capping of the tubes as described, the capsules were ready for use in the in vitro osmosis-driven delayed burst delivery assessment. One tube out of 15 PCL tubes, and one out of 10 xPCL tubes were excluded for being defective, evidenced by immediate release of large quantities of dye upon immersion. This indicates a 92% success rate for the fabrication method.

### 3.3. Phases of Osmosis-Driven Delayed Burst Release

Persistent logging of mass and UV absorbance data enabled construction of charts that depict the mass and dye profiles over the course of the entire experiment (Figure 2). Device mass profiles from both material groups appear split into three phases: an initial uptake phase (1), a post-burst stabilisation phase (2), and a final steady-state release phase (3). The transition between the uptake and post-burst phases occurs at the point in time upon which the device bursts.

In this example, during the initial water uptake phase (1), a steady gain in mass can be noticed concomitant with an absence of dye release. After the moment of burst, and into the post-burst phase (2), the device begins to lose mass, while the dye release sharply increases. In the final steady-state release phase (3), the dye release continues, and mass loss slows.

Devices of all groups display a gradual gain in mass over the initial water uptake period after immersion. After reaching a peak mass gain, a loss of mass occurs, signifying burst of the device. The xPCL devices feature a steep loss in mass over a short period before stabilising and transitioning into a more gradual mass decline. PCL tubes typically lose less mass over a shorter period, before also developing into a more gradual mass loss profile. No dye release was observed prior to devices bursting, with xPCL group devices showing a step-like increase in dye release upon burst followed by a gradual release profile, while the PCL devices lack the burst release and only demonstrate a gradual release of dye after burst. Results relating to the different phases are presented chronologically in the following sub-sections.

### 3.4. The Water Uptake Phase and the Delay to Burst

The water uptake rate was determined from the gradient of the mass gain over the first 14 days after immersion for each tube within a group, where the uptake curves were still close to linear (mean R^2^ = 0.986 ± 0.013, min R^2^ = 0.948). The main parameters evaluated in this phase were water uptake rate, peak mass gained and time to burst which are shown in Figure 3.

The water uptake rate (Figure 3a) generally increases with initial osmogent loading concentration, with statistically significant differences between the 3 and 4 m osmolarity groups of both materials. However, no statistically significant differences were found when comparing the 3 or 4 m groups to the 3.5 m group of each material type. The water uptake rates appear to be similar between device material groups with the same osmolarity (no statistically significant differences found), despite the large difference in wall thickness. The average group peak mass was similar (overall average of 14.4 ± 4.8 mg) between all groups.

The burst delay time of the devices was determined by performing visual, gravimetric, and UV detection methodologies on each device, as depicted in Figure 3b–d, respectively. The burst events occur between 2 points, which corresponds to 2–3 days due to the sampling intervals. Devices across the different groups demonstrated a burst delay time ranging between 12 and 58 days, with 4 m PCL tubes bursting within 16–26 days and 4 m xPCL tubes bursting within 19–26 days, as determined through UV spectrophotometry. Devices with higher osmogent concentration generally burst after a shorter delay, with some 3.5 m concentration group devices bursting within similar times to the 4 m group. For both material types, the variation within a group strongly increases with lower initial loading osmolarity. Detection through mass-based or UV means typically identified device bursts earlier than by eye. The average water uptake rate, peak mass and burst delay of each individual tube and averages within a subgroup are shown in Appendix A.

### 3.5. The Moment of Burst

For dye-based detection of burst, the burst event was regarded to have occurred between the first point that displays a positive detection of dye above baseline and the preceding baseline data point. Therefore, the difference in detected dye between these points was regarded as the dye released at burst. The release of payload immediately after burst was quantitatively determined from the UV data (Figure 4).

The xPCL groups released a considerably larger (and statistically significant) proportion of their payload upon burst in comparison to their PCL counterparts, which is evident from the dye loss values shown in Figure 4c. Comparing averages between groups of the same molarity, xPCL tubes released between 8.1 and 9.2 times more dye than their PCL equivalents. This stark difference was also seen in mass loss. However, only dye release values were used as UV spectrophotometry is more sensitive, and mass loss also depends on the density (hence osmogent concentration) of the released volume. There was no statistical significance found between osmolarities of the same material group.

Representative images of burst sites belonging to both material groups are shown in Figure 4a,b. Images were taken at a single timepoint at the end of the experiment. PCL devices typically demonstrate pinhole bursts or a small fissure (0.6–0.8 mm in length) along the tubular part of the device. While some xPCL devices demonstrate similar pinhole bursts, most display a fissure of greater length (1–5 mm) and width than fissures noticed on PCL devices. No permanent change in overall capsule size or shape was observed for either material or any osmogent concentration.

The release rates were obtained by fitting the gravimetric and UV data over a time interval assessed separately for each group (Figure 5), due to the differences in burst and post-burst timings and behaviours. A time interval was chosen by considering the dye profiles of each tube within a group and determining an interval where the majority of devices were deemed to have entered a steady-state release phase. The extent of dye release by the end of the experiment was calculated by cumulative summation of dye detected at each time point. The total mass released by the end of the experiment was determined by the difference in final mass from the peak mass. After the burst event, mass and dye release were followed until day 72, when the experiment was ended. Post-burst dye release rates show that the xPCL devices were releasing dye approximately twice as fast as the PCL group devices (Figure 5a—*p* = 0.005). The dye release rates also suggest that osmolarity does not determine the extent of release after burst, with release rates within a material-type group being similar.

At day 72, the xPCL devices released more than double the quantity of dye than the PCL devices released at the same stage (Figure 5b—*p* < 0.001). Furthermore, all osmogent concentrations led to a similar total dye release from each material group. Mass loss rates across all groups showed high variability and offer no distinguishable trend. Final mass released by day 72 across all groups was similar, yet with high standard deviations for most groups. A detailed list of post-burst characteristics for each individual tube, and group averages are shown in Appendix A.

The delay to onset of steady-state release indicates that 4 m devices entered a steady state sooner after burst than the 3.5 m devices (8–10 days after peak mass vs. 15–19 days, respectively), with very little difference between PCL and xPCL groups. The 3 m devices from both groups showed a large scatter in burst delay time and behaviour.

### 3.6. Degradability and Cytotoxicity

Full degradability of xPCL and PCL was confirmed through accelerated means by autoclaving discs of both materials in 1 M NaOH (121 °C at pH 14). The alkaline degradation medium with discs made of xPCL turned into a cloudy suspension within 20 min and became clear and free of particles after approximately 30 min, indicating full hydrolysis. For PCL discs, these events happened after 14 and 18 h, respectively.

Cellular metabolic activity was measured spectroscopically by conversion of WST-1 into formazan dye at day 1, 4, and 11 days of cell culturing on glass, PCL and xPCL (extracted and non-extracted) discs. The relative metabolic activity normalised to the glass control group is shown in Figure 6a. Cells cultured on the PCL and xPCL initially showed reduced metabolic activity relative to the control group (*p* = 0.043). However, activity increased over the experimental timeframe with no statistical difference between the means of the material groups compared to the glass control from day 4 onward. There was no statistically significant difference in mean metabolic activity between the PCL and xPCL discs at any time.

Epifluorescence microscopy of the samples revealed that the cells attach to all three surfaces (coverslips, PCL, and xPCL) and the fibroblasts adopt an elongated spindle-like morphology. Consistent with the results from the WST-1 assay, showing increased cell proliferation over time, the images revealed an increase in cell coverage of the sample surfaces over time. Taken together, these preliminary experiments indicate that whilst showing lower initial cell adherence than the glass control, the extracted xPCL does not impede fibroblast proliferation or alter fibroblast morphology, indicating biocompatibility of the material in vitro.

## 4. Discussion

In this work, we assessed osmosis-driven delayed burst release from biodegradable capsules, intended for single-administration vaccination [27,28]. Delayed release of a dye was successfully performed with several improvements over our previous device design, and with new insights gained. Gravimetric and UV spectrophotometry measurements show that all devices follow a multi-stage profile of water uptake, burst, immediate mass loss and payload release, followed by a gradual further loss of mass and payload. Crosslinked polycaprolactone (xPCL) devices released 8–9 times more dye upon burst, followed by a higher steady-state rate of dye release, and larger total release by the end of the experiment, compared to their thermoplastic PCL equivalents. The two main causes for the increased release from xPCL capsules appear to be a larger elastic recovery upon burst, and the formation of larger fissures, from which the payload can be released at a greater rate. The delay time and peak mass at burst were primarily dependent on the osmogent loading concentration through water uptake rate, with no noticeable effect from material type. Utilisation of lower osmogent loading concentrations (3.5 m or 3 m as opposed to 4 m) introduced greater variability in the delay time to burst as well as in release rates of the devices of both material types.

### 4.1. Influence of Material: xPCL vs. PCL

The rates of water uptake between material groups with the same osmogent loading concentration were similar, despite the xPCL tube wall thickness being ~2.6× larger than that for PCL. This implies that the permeability of the xPCL network to water must be essentially 2.6× greater than that of PCL. This could be explained by the lower crystallinity of the network (Appendix A) and the associated increase in free volume [29,30]. In this instance, however, the effect of permeability on the rate of water uptake was counteracted by the thicker walls of the xPCL devices, leading to a similar burst delay time, and facilitating the comparison of release behaviour. The much larger dye release from xPCL devices at burst compared to PCL devices appears to be driven by an elastic recovery of the material. Dynamic mechanical testing of the film samples confirmed that xPCL was indeed more elastic than the PCL material, demonstrating on average less than half the permanent set after deformation. This was also supported by the observation that, for all but one device in the 4 m xPCL group, larger peak mass was associated with greater dye release at burst (Appendix A). Larger peak mass indicates a more inflated capsule, which releases a larger proportion of its volume when punctured and recovering its original shape. The PCL devices did not feature this behaviour at all; here the immediate dye release was similarly small across all osmogent concentrations, with no correlation to peak mass gained. The release of payload upon burst was further facilitated by the xPCL material showing fissures several times larger than those observed at the PCL burst site. However, the size of the burst site appears to be even more influential on the rate of dye release during the steady-state release phases, which were roughly 3–5x higher for xPCL than for PCL.

### 4.2. Effect of Osmogent Concentration

The water uptake rates of the devices show a strong positive correlation with osmogent concentration. Higher initial osmolarity was also associated with a quicker time-to-burst (though not statistically significant), which is expected as each tube of the same material will likely burst at a similar rate of inflation. Burst reproducibility was negatively impacted by lower osmogent concentrations, demonstrated by comparing the burst delay time of 4 m xPCL device at 21.3 ± 2.9 days to 3 m xPCL devices at 28 ± 20 days. This was expected to some extent as material failure is prone to sample-to-sample variation (due to imperfections), which would be exacerbated for lower osmogent concentrations, where the driving force for water uptake and hence, the rate of mechanical loading on the capsule wall would be smaller. However, the magnitude of scatter was larger than expected. While the burst delay time averages display the expected trend, there are samples in the lower osmolarity groups that burst around the same time as, or even sooner than, the 4 m groups. Since all tubes were manufactured and assembled in the same way and being assigned to a specific osmolarity group at random, there is currently no explanation for the exceedingly large variance in the 3 m xPCL group. It is noted though, that future experiments would benefit from larger sample populations. Due to the increase in burst variability at lower glucose concentrations, it would be advised to maintain a 4 m glucose concentration or greater, and use other variables such as material water permeability, wall thickness and the extensibility of the material to prolong or tune the burst delay time. In our previous work, PCL (CAPA 6500, Perstorp, molecular weight 50 kg/mol) tubes that were loaded with a 5 molal glucose solution burst after 8.7 ± 2.9 days.

### 4.3. Evaluation of Methods

In this study, several methods for monitoring the device behaviour were employed in parallel. Gravimetric measurements were particularly useful during the water uptake phase, when no dye had been released yet that could be detected. Regular weighing enabled calculation of the water uptake rate and peak mass gained, which helped to confirm the moment of burst. UV and gravimetric analysis on average detected bursts 3.5 ± 1.7 days earlier than visual identification, demonstrating inadequate sensitivity of the human eye for this purpose. However, it should be noted that the xPCL tubes showed a much closer agreement between visual and UV detection than the PCL tubes, which is likely due to the burst behaviour causing a greater and more immediate colour change. Mass recordings were less informative after burst, for a number of reasons: (a) it depends on density, which is different for the different osmogent loading concentrations; (b) it is less sensitive than UV spectrophotometry, and (c) mass loss upon release is counteracted by ongoing osmosis-driven water uptake. These reasons limit the usefulness of mass loss as a method of release analysis. It is therefore not surprising that the correlation between mass loss and dye release was rather poor, and we recommend using the latter to assess the devices’ release behaviour.

The sampling time of 2–3 days employed may have contributed to the UV and gravimetric data spread of all the tube groups. This especially holds for the 4 m groups, where standard deviations in burst times across both material groups fall below 3 days. Therefore, it is possible that certain tubes burst on day 1 of that interval that were not detected until a further 2 days later. Ideally, in future studies assessments would be made daily or more frequently by utilising automated means. Particularly for devices constructed from flexible or elastomeric materials that show a greater bolus release at the time of burst, time lapse imaging might be sensitive enough to determine the burst delay times of large quantities of samples, in a less labour-intensive way. Gravimetric measurements would still be recommended during the water uptake phase to provide useful information, whereas UV spectrophotometry at strategic moments would provide adequate information on payload release rate.

### 4.4. Degradation and Cytocompatibility

Due to the slow speed at which PCL degrades under physiological conditions, we utilised an accelerated approach similar to Jansen et al., featuring an alkaline medium and high temperature [31]. Such a combination vastly reduces the degradation time of the thermoplastic PCL from 2–4 years to under 20 h. Use of pH-mediated methods to accelerate hydrolysis of biodegradable polymers is a common approach [32]. While there is no published degradation data for this particular crosslinked PCL network, we assume it will be similar to high-molecular-weight thermoplastic PCL. This viewpoint has been assisted from an on-going long-term study under physiological conditions investigating xPCL discs composed from crosslinking of 10 kg/mol macromer. After approximately 2.5 years these disks lost on average 15 ± 1.7% (*n* = 6) of their starting mass, but the structural integrity of the network appears disrupted as the disks are friable during gentle manipulation with tweezers or by hand. Degradability is an advantage of this device, eliminating the need for surgical removal after the device has fulfilled its function.

Cytotoxicity is an important aspect in the evaluation of novel biomaterials. Given the intention to implant these capsules subcutaneously, we assessed cellular behaviour of human dermal fibroblasts in the presence of xPCL, compared to medical grade PCL (a similar material that is in clinical use) and a glass slide control. The attachment of cells to substrates is mediated by the adsorption of proteins from culture medium. The quantities and ratio of proteins adsorbing onto a substrate strongly depends on its surface chemistry, most notably wettability [33]. This may explain the lower initial cell adherence to both PCL materials compared to glass, as was observed both microscopically and in a metabolic activity assay. This may have been exacerbated by the seeding method. One hour after seeding a droplet of cell suspension onto the surfaces, the well was filled with culture medium, which may have flushed off a larger number of non- or weakly adherent cells from the PCL materials than from the glass. However, from day 4 there was no statistical difference between the metabolic activity on any of the substrates, and the microscopic images showed highly similar cell coverage and morphology on day 11. This indicates that despite a lower seeding density, the fibroblasts on the PCL and xPCL were able to proliferate and essentially catch-up with the control group. Overall, these are encouraging preliminary results, and in accordance with previous work on different polymers [34].

PCL oligomers are largely accepted to constitute an intrinsically biocompatible system when interacting with various cell and tissue types [35]. Moreover, our findings are in agreement with previous literature that investigated the cytotoxicity of photo-crosslinked PCL networks using similar methods [36,37]. Upon implantation of any biomaterial, a foreign body response is to be expected. Although the foreign body response to PCL is mild [35], ultimately the capsules developed herein will have to be investigated in vivo to confirm their safety for the intended purpose of vaccine delivery. While each process component has not been individually experimentally assessed here for cytotoxicity, steps had been taken to minimise the risk of residual toxicity of resin components. Benzyl alcohol is not currently classified under ICH residual solvent guidelines, but is often used as an excipient in intramuscular, intravenous, and topical medicinal products, as well as in the food and cosmetics industries [38,39]. TPO is a widely used photo-initiator and various (extracted) crosslinked networks formed from photo-curable resins containing TPO have been shown to have good cytocompatibility from standard in vitro tests [34,36,40]. Finally, thorough purification methods were performed resulting in extracted photo-cured networks, that can be considered non-toxic when compared against thermoplastic medical grade PCL or the glass control group.

### 4.5. Outlook

This study demonstrates several significant improvements over our proof-of-principle paper [17], both in methodology and in the material choice. As discussed in that study, we still consider this a potentially powerful platform for single-administration vaccination, as in some cases the increased inconvenience of the subcutaneously inserted capsule will outweigh the difficulties and cost associated with repeated administrations, and prove more practical or reliable than other technologies currently being developed. Further improvements are necessary before these capsules can be deployed, most notably a further increase in release of payload at the moment of burst. We envisage this to be achieved through employing a fully amorphous elastomer as the tube material to guarantee full shape recovery, combined with higher elongation at break to allow for a larger peak mass, and hence, larger burst release. Studies using such elastomers are ongoing, in which the influence of crosslink density on peak mass and time to burst are being investigated. Some studies have suggested that alternative dosing regimens may illicit a stronger immune response than repeated bolus injections. Research into escalating dose [41], sustained release, and pulsatile release methods [42] has been performed in rodents, whilst a recent study concluded that slow-release methods led to improved immune responses in non-human primates [43]. Therefore, considerations regarding the optimal release profile of antigenic material will need to be continually assessed. Yet, the repeated bolus pattern has shown great clinical success and can be expected to remain the predominant regime for many years to come. Our aim is to facilitate the deployment of the repeated bolus regime in specific, challenging settings, through the delayed burst mechanism of our capsules. However, it would be possible for us to adjust the capsule design into a degradable sustained-release osmotic pump, if further supporting research indicates the immunogenic benefits of such an approach.

Other steps towards translation are being considered in parallel, which include automation of production, quality control, safety and methods of administration. The production failure rate of only 12% achieved here, and ability to quickly identify defective devices are an attribute to the suitability for translation, along with the automated dip coating setup which manufactures 10 devices simultaneously without operator intervention. Encouraging results were obtained in this study with respect to degradability and cytocompatibility, and both will be taken further by identifying and investigating the safety of degradation products, and moving towards tissue response in vivo. Administration of the devices is envisaged through the use of a plunger-style applicator mounted on a 12–14 Gauge needle. This is similar to the method used for the female contraceptive implants such as Implanon™ in wide use today. We envisage our devices being implanted at the time of the initial injection, where the device is used to store and deliver a timely booster dose of vaccine at a later date without the need to return to a clinic or to be visited by a health care professional.

In our proof-of-principle paper, we provided a detailed comparison between our capsules and other approaches for single-administration vaccination [17]. In short, the advantages over other approaches were the potential for bolus profile, ability to control the delay time, the manufacture being independent of the payload (protecting antigens from harsh conditions), and the possibility to remove the implant if necessary. In particular settings, such as vaccination of livestock and wildlife or developing world vaccination programs where logistical issues can be extreme, the utility of these devices will outweigh the more invasive administration procedure [17]. A further advantage is that the payload is independent from the delivery device, and therefore other payloads (i.e., hormones, cellular therapies, etc.) could potentially be rapidly deployed using the system. Appropriate considerations would have to be made regarding the storage of other payloads during the delay period, and the suitability of the nature of the burst of the devices for that application. Since that study five years ago, research into biomaterials-based solutions for single-administration vaccination has been ongoing, predominantly in the form of microparticles [13,14], and microneedle arrays [15,16]. In that time, the use of osmosis as a trigger for delayed burst release has remained relatively unexplored, with only a conceptual demonstration of such release from PLGA spheres [44]. In that study, hollow spheres with radii of 1.5–2.2 mm and wall thicknesses of 50–100 µm were formed by dip coating spheres of ice mounted on a needle into a PLGA/DCM solution. The dried spheres were then filled with a saturated salt solution of NaCl or LiCl, before being sealed with a drop of PLGA/DCM solution. In vitro testing indicated that the delay time could be adjusted between approximately 1.5–17 days by changing the osmogent solution, capsule radius, and shell thickness. It was shown that beyond 10 days, the spheres were subject to significant degradation, which resulted in early bursts. No implantation method was discussed.

## 5. Conclusions

We successfully demonstrated delayed burst release of a dye from devices manufactured from a resorbable, non-toxic polymer. The introduction of crosslinks to polycaprolactone did not generally influence the delay to burst time, but did affect the characteristics of the payload release after burst. The more elastic crosslinked polycaprolactone (xPCL) devices displayed up to 8–9× greater initial release immediately following burst by releasing approximately 9% of total loaded dye, compared with ~1% from non-crosslinked polycaprolactone (PCL) devices. Additionally, by day 72, the xPCL devices had released up to 2–3 times more payload than comparable PCL devices. However, even for xPCL devices this was still less than 40% of the total payload. These findings suggest that a more elastic material could further approximate the bolus release profile of current booster shots. The delay to burst could be extended by reduction in the concentration of glucose that was loaded into the device during manufacture. This was attributed to the rate of water uptake into the device, with faster water uptake occurring when the device was loaded with higher concentrations of glucose. However, the delay to burst time was most reproducible when glucose-loading concentration was highest, so other mechanisms to tune the burst delay time should be explored.

## Figures and Tables

**Figure 1 pharmaceutics-13-00434-f001:**
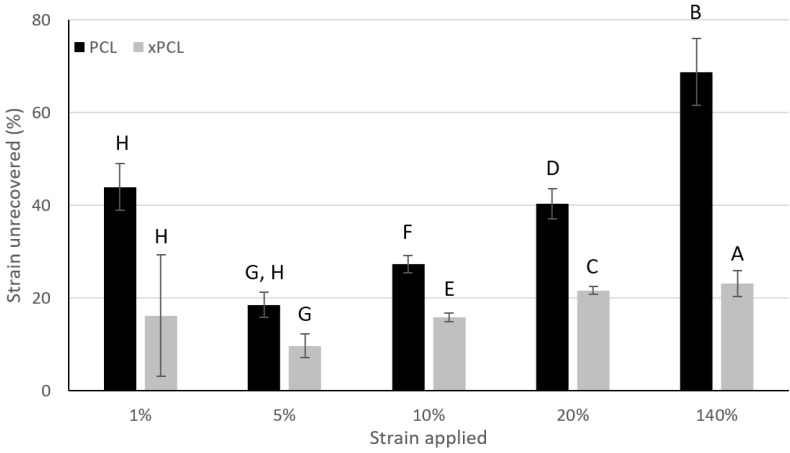
Comparison of the average permanent set following each target strain exerted. Data are normalised against the strain applied. Recovery time after each strain was 15 min. Error bars indicate standard deviation. Groups that do not share a letter are statistically different from one another (*p* < 0.05). PCL *n* = 5, xPCL *n* = 4.

**Figure 2 pharmaceutics-13-00434-f002:**
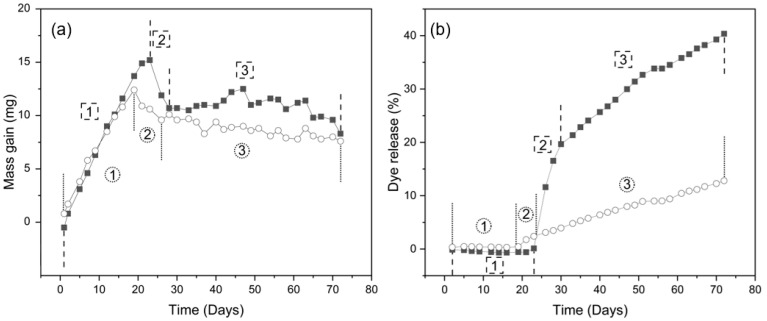
Phases of osmosis-driven delayed burst release, demonstrated through mass change (**a**) and dye release (**b**) profiles of one representative capsule each of PCL (○) and xPCL (■) 3.5 m devices. Phases are shown as (1) water uptake, (2) post-burst stabilisation, and (3) steady-state release. The point of burst is depicted by the dashed lines that separate phases 1 and 2.

**Figure 3 pharmaceutics-13-00434-f003:**
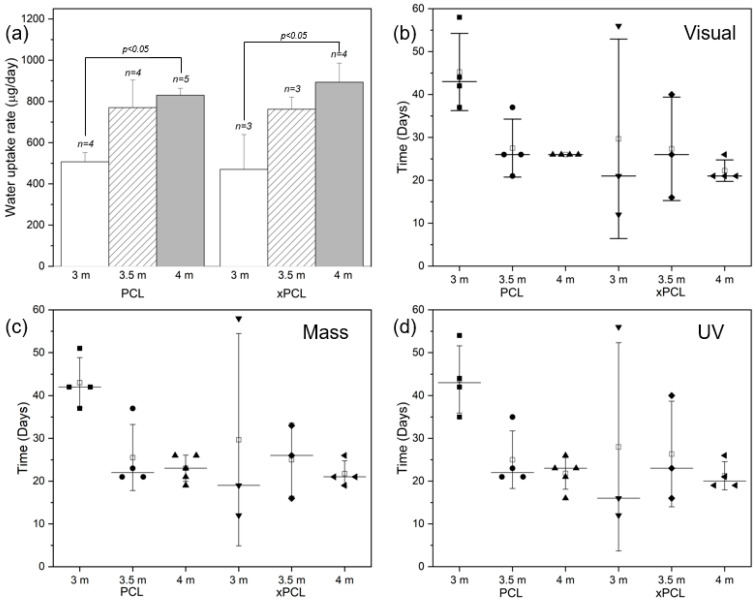
Characteristics of the water uptake phase, grouped by material type and initial osmogent loading concentration. (**a**) Water uptake rates (average ± standard deviation; population values are indicated above each bar). Scatterplots displaying the burst delay time of the devices determined by (**b**) visual detection, (**c**) mass-based detection, and (**d**) UV spectrophotometry detection. Each point represents an individual device, whiskers represent standard deviation, horizontal line indicates the median, and the open box displays the mean.

**Figure 4 pharmaceutics-13-00434-f004:**
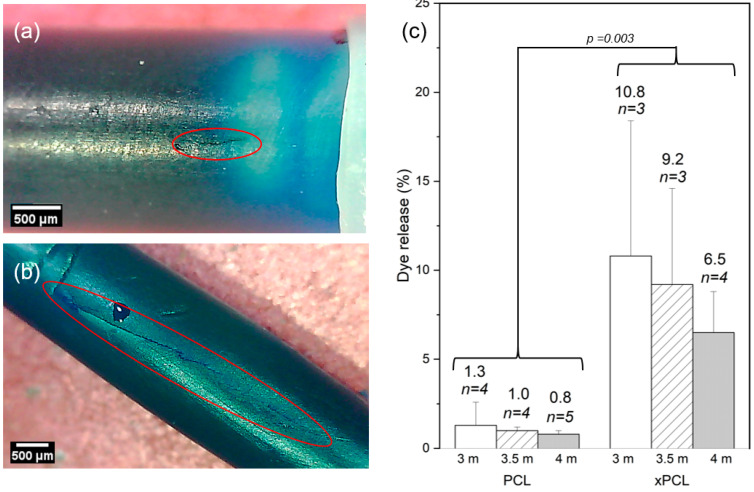
Moment of burst. Microscope images of representative burst sites of a PCL tube (**a**) and a xPCL tube (**b**). Red ovals are to highlight the lengths of the burst areas. Fissure lengths circa 0.7 mm (**a**), and 5 mm (**b**). (**c**) Average percentage of loaded dye released upon burst of the devices. Error bars represent standard deviation. Population sizes are indicated above each bar along with the %-age of dye released.

**Figure 5 pharmaceutics-13-00434-f005:**
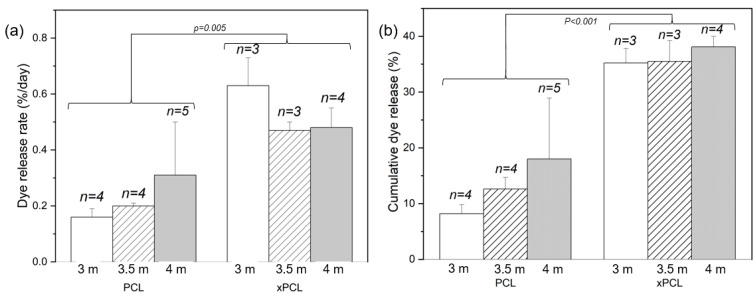
Post-burst release. Average release rates (**a**) and cumulative release by day 72 (**b**) of dye from devices of the three osmolarity groups and each material type. Error bars indicate standard deviation. Population sizes are indicated above each bar.

**Figure 6 pharmaceutics-13-00434-f006:**
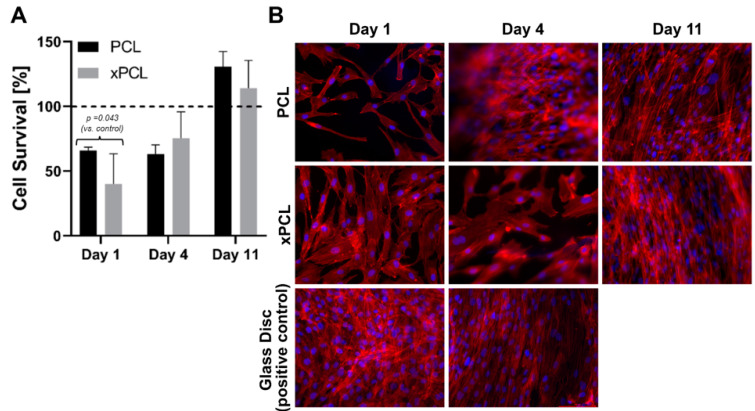
(**A**) WST-1 assay determined average metabolic activity over 14 days of culturing in relation to control group (glass discs). Data represent an average of triplicate samples ± SD. Reference line at 100% indicates glass disc signal. (**B**) Representative microscope images of DAPI/phalloidin-stained fibroblasts cultured on top of, PCL, xPCL, and glass discs, after 1, 4, and 11 days of culturing. Magnification 20×.

## Data Availability

Data have been provided within the Appendix A.

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
