# Peer review of "Elastic Bioresorbable Polymeric Capsules for Osmosis-Driven Delayed Burst Delivery of Vaccines"

_pharmaceutics, 2021, doi:10.3390/pharmaceutics13030434_

Round 1

Reviewer 1 Report

This manuscript studied the feasibility of a potential concept of using Elastic bioresorbable polymeric capsules for delayed burst release of vaccines using the osmosis principle. It is a nicely written and well-organized manuscript, and below are some of the comments to be addressed.

Comments:

  • The concept is centered around the feasibility of osmosis-driven delayed “burst” release, however, based on Figure 2B both capsules showed delayed “sustained release” rather than “burst release”. The studied capsules behaved more likely as delayed “sustained” release capsules than delayed “burst release” systems. xPCL showed ~20% burst and the remaining 80% dye release is sustained over a period of days/months, and PCL did not show any burst phase and showed sustained release profile.
    • The key question is: How does the vaccine delivery target profile looks like (ie., what is the target release profile for vaccine delivery) – do they need to be released for an extended period of time (OR) they need a delayed phase where there is no drug release for a time and then all the drug is released at once (Burst)? Adding additional explanation in the discussion on these concepts will be helpful to the readers in understanding how the profiles seen with the developed xPCL and PCL capsules can fit the need.
  • In figure 4. Authors showed fissure dimensions, has authors compared multiple samples for fissure dimensions – imaging studies of the samples over time? How uniform are the fissures generated for capsules (range)? Since the dye release was more variable in xPCL samples, how much this could be a concern when the capsules are loaded with a potent vaccine formulation and if the release is inconsistent from one sample to another.
  • The addition of basic polymer structure xPCL vs PCL would be helpful
  • Please comment in the manuscript on the safety of each process material (for example photo-initiator, Methacrylic anhydride) used during manufacturing. Are any of them used in clinical products?
  • Is there any approximate estimation of the cross-linked polymer molecular weight?
  • “Poly(ε-caprolactone) (PCL) macromonomers of high purity and quantitative hydroxyl-to-methacrylate conversion were obtained as confirmed by 1H-NMR spectrometry (Supplementary Figure S1)” – Other than NMR are there any characterization studies performed to test the presence of unreacted materials (%), what is the approximate percent purity of the synthesized material?

Author Response

We’d like to thank all reviewers for their time and suggestions which have improved our manuscript.

Reviewer 1

  1. “The key question is: How does the vaccine delivery target profile looks like (ie., what is the target release profile for vaccine delivery) – do they need to be released for an extended period of time (OR) they need a delayed phase where there is no drug release for a time and then all the drug is released at once (Burst)? Adding additional explanation in the discussion on these concepts will be helpful to the readers in understanding how the profiles seen with the developed xPCL and PCL capsules can fit the need.”

Response:

Indeed, this is a key question for the broader research field. We have added a section in the Discussion including references to 3 studies into release regimes, and discussed the role of our research in this respect. Lines 604-615.

“Some studies have suggested that alternative dosing regimens may illicit a stronger immune response than repeated bolus injections. Research into escalating dose [41], sustained release, and pulsatile release methods [42] has been performed in rodents, whilst a recent study concluded that slow-release methods led to improved immune responses in non-human primates [43]. Therefore, considerations regarding the optimal release profile of antigenic material will need to be continually assessed. Yet, the repeated bolus pattern has shown great clinical success and can be expected to remain the predominant regime for many years to come. Our aim is to facilitate the deployment of the repeated bolus regime in specific, challenging settings, through the delayed burst mechanism of our capsules. However, it would be possible for us to adjust the capsule design into a degradable sustained-release osmotic pump, if further supporting research indicates the immunogenic benefits of such an approach.” 

  1. “In figure 4. Authors showed fissure dimensions, has authors compared multiple samples for fissure dimensions – imaging studies of the samples over time? How uniform are the fissures generated for capsules (range)? Since the dye release was more variable in xPCL samples, how much this could be a concern when the capsules are loaded with a potent vaccine formulation and if the release is inconsistent from one sample to another.”

Response:

Fissure/crack dimension ranges are now included within the Results section, adjacent to previous mentions of general size (lines 384 and 386). Also, a statement of the images being taken at a single timepoint at the end of the experiment was added (Line 383/384).

We aimed to minimise variability in this work, and believe this can be further minimised in the future by moving to fully amorphous elastomers. The potential impact of variability in potential vaccine release profiles on effectiveness links to the question of optimal release profile discussed above; it remains a key question for the broader field.

  1. “The addition of basic polymer structure xPCL vs PCL would be helpful”

Response:

A new figure has been added to the Supplementary Data (Figure S2) showing the basic polymer/macromer structures, and is referred to in the Results section – Line 269/271.

  1. “Please comment in the manuscript on the safety of each process material (for example photo-initiator, Methacrylic anhydride) used during manufacturing. Are any of them used in clinical products?”

Response:

A new paragraph commenting on purification methods and safety data regarding the resin components has been added from line 578:

“While each process component has not been individually experimentally assessed here for cytotoxicity, steps had been taken to minimise the risk of residual toxicity of resin components. Benzyl alcohol is not currently classified under ICH residual solvent guidelines, but is often used as an excipient in intramuscular, intravenous, and topical medicinal products, as well as in the food and cosmetics industries [38,39]. TPO is a widely used photo-initiator and various (extracted) crosslinked networks  formed from photo-curable resins containing TPO have been shown to have good cytocompatibility from standard in vitro tests [34,36,40].  Finally, thorough purification methods were performed, resulting in extracted photo-cured networks that can be considered non-toxic when compared against thermoplastic medical grade PCL or the glass control group.”

  1. “Is there any approximate estimation of the cross-linked polymer molecular weight?”

Response:

The crosslinked network molecular weight is essentially infinite, as every macromer unit will be covalently bonded to at least one other macromer unit, making the cured structure “one-piece”.

We have added a sentence saying this explicitly – Line No: 267/268          

the crosslinked network has an essentially infinite molecular weight”

For clarity, we have added “thermoplastic” adjacent to discussion of PCL in methods (Line 156), and discussion (Line 455) to emphasise the non-crosslinked nature of this material.

  1. ““Poly(ε-caprolactone) (PCL) macromonomers of high purity and quantitative hydroxyl-to-methacrylate conversion were obtained as confirmed by 1H-NMR spectrometry (Supplementary Figure S1)” – Other than NMR are there any characterization studies performed to test the presence of unreacted materials (%), what is the approximate percent purity of the synthesized material?”

Response:

Only 1H-NMR was performed. We have calculated the relative amount of methacrylic anhydride (the only impurity in the 1H-NMR spectra) to be 0.1wt% (mentioned in the legend of Supplementary Figure S1), implying a macromer purity of 99.9 % (stated in the Results section) Line 263.

Reviewer 2 Report

The researchers in this study have produced crosslinked PCL  devices by dip coating methacrylated PCL oligomers and photocuring. Their release behaviour was assessed in comparison to devices prepared from non-crosslinked PCL. Their work could be considered as a potentially powerful platform for the single administration vaccination. In vivo experiments would be interesting in a following stage.

This reviewer's recommendation is acceptance of the manuscript in present form.

Author Response

Reviewer 2

  1. “The researchers in this study have produced crosslinked PCL devices by dip coating methacrylated PCL oligomers and photocuring. Their release behaviour was assessed in comparison to devices prepared from non-crosslinked PCL. Their work could be considered as a potentially powerful platform for the single administration vaccination. In vivo experiments would be interesting in a following stage.

This reviewer's recommendation is acceptance of the manuscript in present form.”

Response:

Thank you for your supportive comments on the manuscript and your enthusiasm in our work!

Reviewer 3 Report

This manuscript by Samson et al., describes efforts to generate vaccine delivery devices for implantation. These devices work by osmosis-driven bursting of the device, releasing the contents in a somewhat timed manner. The work described here tested devices made from a polymer (xPCL) and compared to PCL, with the hypothesis that xPCL would have more elasticity and improve on previous work done with PCL. 

Although this work is outside of my direct field (I am a vaccinologist) I found the work to be interesting and the studies done appropriate for the hypotheses being tested. The manuscript is well written and the introduction was very clear for explaining the goals of the study and the background rationale.

I have 3 main concerns that I would like addressed prior to acceptance of the manuscript.

  1. The figures showing quantitative data all need to have statistical analysis added. Particularly in the cases where a conclusion are being made for one of the materials releasing dye faster or at greater quantity. All figures should be reevaluated and statistical analysis added. 
  2. For Figure 6, the authors conclude that the PCL and xPCL do not impede fibroblast proliferation or alter morphology, and yet it appears that there is a difference between PCL and xPCL compared to the inert control material (glass). The authors make a point later that the PCL is already used/being tested clinically in humans, so maybe that is the comparison they are making--that PCL and xPCL are not different and since PCL is ok in humans that xPCL should be fine too? Regardless--the conclusion as currently written is not supported by the data. 
  3. The authors indicate that the polymers take years to degrade under normal physiological conditions. How then would the authors imagine these devices being administered to humans or animals? Would the devices be left in the individual? I would like to see more extensive discussion of the practical application of the work and how the authors foresee this working. The authors tend to gloss over this point by stating line 570-575 "In short, the advantages over other approaches were the potential for bolus profile, ability to control the delay time, the manufacture being independent of the payload (protecting antigens from harsh conditions), and the possibility to remove the implant if necessary. In particular settings, these will outweigh the more invasive administration procedure." In what settings? Also--could this be used for drug delivery other than vaccines? 

Overall I think this is an interesting approach for drug or vaccine delivery and look forward to seeing the final manuscript!

Author Response

We’d like to thank all reviewers for their time and suggestions which have improved our manuscript.

Reviewer 3

  1. “The figures showing quantitative data all need to have statistical analysis added. Particularly in the cases where a conclusion are being made for one of the materials releasing dye faster or at greater quantity. All figures should be reevaluated and statistical analysis added.”

Response:

Statistical analyses have been performed and were found to support the previously drawn conclusions. All numerical based figures and evaluative text have been updated to reflect the results of the statistical analysis. Section (2.7 – Line 249 onwards) added to methodology explains which tests were used for which figures, and how to interpret the presented results of the tests.

  1. “For Figure 6, the authors conclude that the PCL and xPCL do not impede fibroblast proliferation or alter morphology, and yet it appears that there is a difference between PCL and xPCL compared to the inert control material (glass). The authors make a point later that the PCL is already used/being tested clinically in humans, so maybe that is the comparison they are making--that PCL and xPCL are not different and since PCL is ok in humans that xPCL should be fine too? Regardless--the conclusion as currently written is not supported by the data.”

Response:

We have elaborated on our discussion of the cell experiment (Line 558-571), explaining the lower initial number of adhered cells:

“The attachment of cells to substrates is mediated by the adsorption of proteins from culture medium. The amounts and ratio of proteins adsorbing onto a substrate strongly depends on its surface chemistry, most notably wettability [33]. This may explain the lower initial cell adherence to both PCL materials compared to glass, as was observed both microscopically and in a metabolic activity assay. This may have been exacerbated by the seeding method. One hour after seeding a droplet of cell suspension onto the surfaces, the well was filled with culture medium, which may have flushed off a larger number of non- or weakly adherent cells from the PCL materials than from the glass. However, from day 4 there was no statistical difference between the metabolic activity on any of the substrates, and the microscopic images showed highly similar cell coverage and morphology on day 11. This indicates that despite a lower seeding density, the fibroblasts on the PCL and xPCL were able to proliferate and essentially catch-up with the control group. Overall, these are encouraging preliminary results, and in accordance with previous work on different polymers [34].”

As the change in metabolic activity with time of cells culture on the PCL materials compared to those on the glass control either remains the same or is higher, this means proliferation is not impeded.

  1. “The authors indicate that the polymers take years to degrade under normal physiological conditions. How then would the authors imagine these devices being administered to humans or animals? Would the devices be left in the individual? I would like to see more extensive discussion of the practical application of the work and how the authors foresee this working. The authors tend to gloss over this point by stating line 570-575 "In short, the advantages over other approaches were the potential for bolus profile, ability to control the delay time, the manufacture being independent of the payload (protecting antigens from harsh conditions), and the possibility to remove the implant if necessary. In particular settings, these will outweigh the more invasive administration procedure." In what settings? Also--could this be used for drug delivery other than vaccines?”

Response:

Added information regarding the anticipated administration method – Line 627-630.

We expect to use a plunger-style applicator similar to what is used with Implanon™ and other female contraceptive implants that share a similar geometry and scale of our capsules.

Added a statement referring to the benefit of biodegradability/bioresorbability after the device has served its purpose, where no surgical removal is required. Line 553/554.

After including only a short summary of the extensive discussion of the practical application from our earlier publication with reference [17], we have now expanded this section to be more comprehensive in this manuscript. Line 635-638.

Added more specific details on what settings we feel our device would offer the greatest potential benefit. Line 636-638

Added a clear statement mentioning the potential use for other payloads and the flexibility to adapt to delivery of such alternative payloads due to the independence of the delivery system and burst ‘mechanism’ from the payload itself. Line 638-643.